# Hierarchical abstraction drives human-like 3-D shape processing in deep learning models

Shuhao Fu[1]*, Philip J. Kellman[1], Hongjing Lu[1,2]

**1** Department of Psychology, University of California Los Angeles, Los Angeles, California, United States of America, **2** Department of Statistics, University of California Los Angeles, Los Angeles, California, United States of America

* fushuhao@g.ucla.edu

## Abstract

Both humans and deep learning models can recognize objects from 3D shapes depicted with sparse visual information, such as a set of points randomly sampled from the surfaces of 3D objects (termed a point cloud). Although deep learning models achieve human-like performance in recognizing objects from 3D shapes, it remains unclear whether these models develop 3D shape representations similar to those used by human vision for object recognition. Evidence suggests that training with approximately 10,000 object instances enables models to acquire representations of local geometric structures in 3D shapes. We hypothesize, however, that their representations of 3D global shapes are still limited. To test this hypothesis, we conducted three human experiments systematically manipulating point density and object orientation (Experiment 1), local geometric structure (Experiment 2), and part configuration (Experiment 3). Human performance was stable across conditions in the first two experiments, but declined significantly in the part-scrambled condition of the final experiment. We compared human performance with two types of deep learning architectures: convolution-based models (e.g., DGCNN) and transformer-based models (e.g., Point Transformer). The transformer-based models more closely captured human performance patterns across experimental conditions. Ablation simulations revealed that this advantage is largely driven by progressive downsampling operations that enable hierarchical abstraction of 3D shapes.

## Author summary

Humans can recognize 3D objects at a glance, even when they are depicted only as sparse sets of dots sampled from their surfaces, known as point clouds. We asked whether modern deep learning systems rely on the same kind of shape representations as humans or achieve recognition in different ways. This study combined human experiments with model evaluations. Participants viewed point cloud objects while we made recognition progressively more difficult by

**Data availability statement:** All human data, statistical analysis code, network training and testing code are available on Github at https://github.com/fushuhao6/hierarchical_abstraction_in_3D_object_recognition.git. All stimuli used in experiments are available on Zenodo at link https://doi.org/10.5281/zenodo.17158227.

**Funding:** This work was supported by the National Science Foundation under Grant BCS-2142269 (A Unified Theory for Perception of Physical and Social Dynamics, https://www.nsf.gov/awardsearch/showAward?AWD_ID=2142269&HistoricalAwards=false) to HL. The funders had no role in study design, data collection and analysis, decision to publish, or preparation of the manuscript.

**Competing interests:** The authors have declared that no competing interests exist.

reducing the number of dots, flipping objects upside down, distorting local geometric properties, or scrambling parts into new configurations. Humans remained highly accurate in most cases but struggled when the part configuration was disrupted, highlighting a strong dependence on global 3D shape. We then compared two leading deep learning models, and identified the critical computational components responsible for achieving human-like performance. Our results showed that progressive downsampling, which constructs increasingly abstract shape representations, is the primary factor underlying human-like robustness, whereas attention mechanisms contribute only secondarily.

## Introduction

Objects in the natural world possess physical properties such as geometric shape, volume, and material composition. The human visual system is highly efficient at extracting these properties from visual input, often within a brief glance at an image. Among the various object attributes, the ability to perceive and recognize three-dimensional (3D) shape is widely regarded as fundamental for everyday behaviors such as navigation, object manipulation, and interaction with the external environment. A substantial body of research [1–5] has demonstrated that human object recognition does not merely rely on memorizing collections of two-dimensional (2D) retinal images across viewpoints. Instead, humans construct internal 3D representations of objects that support robust recognition. 3D object representations provide an object-centered description of global shape by encoding the spatial relations among an object's features, often referred to as a structural description. When this structural information is available in the visual input, perception based on global shape remains robust to variations in object appearance arising from changes in viewpoint, occlusion, illumination, and other imaging conditions.

The importance of 3D shape perception is further underscored by its early emergence in development. Sensitivity to 3D structure is found in human infancy [6], suggesting that mechanisms for representing 3D shape from visual input are present early in life. As development progresses, toddlers between 18 and 24 months exhibit a pronounced "shape bias" in word learning, increasingly generalizing object names based on shape rather than texture, color, or other perceptual features as their vocabularies expand [7]. Although sensitivity to texture and other visual cues continues to mature, these features become secondary to shape in guiding object recognition and naming objects. Notably, even when texture information is entirely absent from the visual input, humans are still capable of accurately recognizing objects based solely on their 3D geometric structure.

A striking demonstration of this capacity is human recognition of objects presented as point clouds, consisting of discrete points sampled along object surfaces (see Fig 1). Despite the absence of continuous contours, shading, and texture, humans readily recognize 3D objects from such minimal visual information [8–12]. These behavioral findings highlight the robustness and flexibility of human

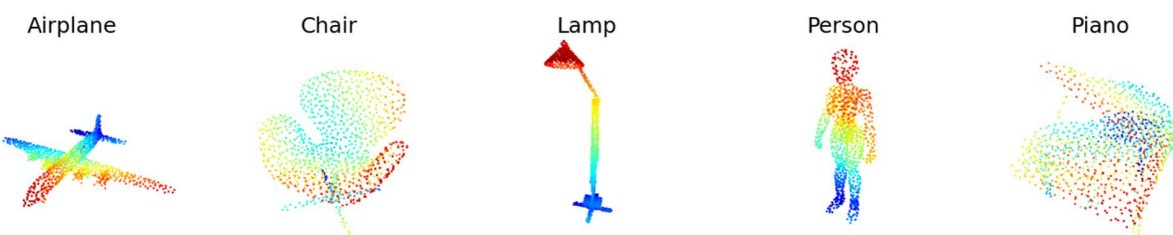

**Fig 1. Example stimuli used in the experiment.** Each object is visualized as a sparse point cloud sampled from its 3D surface. Colors represent depth, with red indicating proximity and blue indicating distance. In the experiments, point clouds were displayed in black and presented as rotating GIFs.

3D shape perception. Meanwhile, neuroscience research indicates that shape representations emerge through a hierarchical sequence of processing stages along the ventral visual pathway in the brain. This pathway extends from posterior occipitotemporal regions of the inferior temporal cortex (IT), including the lateral occipital cortex (LO), to more anterior regions encompassing the fusiform gyrus. Object-related information is ultimately transmitted to the anterior temporal lobe (ATL), where it is integrated into multimodal semantic representations of objects [13,14]. While prior research has emphasized the contribution of the dorsal pathway to object recognition, more recent findings point to a more sophisticated, distributed network underlying global shape processing. The ventral pathway appears to be specialized for extracting local shape features and supporting object recognition based on these features. However, recognition of objects defined by global shape depends on interactions between the dorsal and ventral pathways, which together facilitate the formation of 3D object representations that are robust to variations in viewpoint and occlusion [15].

In parallel with advances in behavioral and neuroscience research on 3D shape perception, recent progress in deep learning has led to the development of specialized architectures for object recognition from 3D point clouds. Two major classes of models have emerged. The first class includes graph-based architectures such as Dynamic Graph CNN (DGCNN), which extend graph neural network approaches by dynamically constructing local neighborhoods in feature space to learn local geometric structure [16]. Earlier models such as PointNet introduced foundational methods by learning spatial features from raw 3D data of point clouds [17]. DGCNN-based models have demonstrated human-like performance across a range of 3D object recognition tasks.

The second class comprises transformer-based architectures adapted for 3D data of point clouds, including the Point Transformer [18], which leverage self-attention mechanisms to capture relations among points. These transformer-based models likewise achieve human-level performance in 3D object recognition. Despite their comparable behavioral performance, it remains unclear whether these different model classes develop internal representations of 3D shape that are similar to, or distinct from, those employed by humans.

Previous research on image-based object recognition has shown that pre-trained deep convolutional neural networks (DCNNs) [19–21] can acquire internal representations that differ from those of humans, despite exhibiting similar behavioral performance in typical testing conditions. For example, Baker et al. [22] found that DCNNs struggled to classify objects in images based on their 2D global shape. In one experiment, they presented CNNs with objects that preserved the 2D global shape but were filled with textures from other objects. The networks showed a strong bias for classifying based on textures rather than shapes. Further experiments revealed that CNNs could not reliably classify objects based on outlines alone, indicating a reliance on local features rather than global shapes. In a separate experiment, these investigators found that, using silhouettes that networks could correctly classify, the addition of minute serrations along the bounding contours reduced network classification to below chance performance. In contrast, human object classification

was not disrupted by this manipulation. These findings suggest that while CNNs can access local shape features in images, they do not form global shape representations crucial for human-like object recognition ([23], see a review [24]).

To address whether deep learning models trained to recognize 3D objects from point clouds acquire representations similar to or different from those of humans, it is necessary to combine systematic experimentation with model ablation and intervention approaches to identify the core computational components underlying 3D shape recognition in both humans and models. In this paper we compared 3D object recognition in humans and deep learning models. Through a set of experimental manipulations, we analyzed how both humans and models recognize 3D objects, particularly in challenging conditions where local and global shape features were manipulated. This included disrupting local geometric properties, presenting objects from unusual viewpoints, and varying the global configuration of object parts. We then tested the two types of deep learning models using the same tasks, and conducted ablation/intervention studies to identify the core computational mechanisms underlying 3D object recognition in the DNN models.

## Modeling methods

### Dataset

We used stimuli selected from the publicly available point cloud dataset, ModelNet40 [25]. The ModelNet40 dataset includes a set of 3D CAD models from 40 object categories. There are 12,311 3D CAD objects in total, with 9,843 objects used for training and 2,468 for testing. The point clouds consist of 1,024 points sampled uniformly from the surface of each 3D CAD model.

### Deep learning models

We evaluated two deep learning models for 3D object recognition from point cloud data: a convolution-based model (DGCNN; [16]) and a transformer-based model (Point Transformer; [18]). Both models take 3D coordinates of points sampled from a 3D shape and generate feature embeddings for object classification. A more detailed illustration of the model architecture is shown in Fig 2 and a summary comparison between the two models is in Table 1.

DGCNN processes point clouds as graphs using EdgeConv layers, which extract local geometric features by comparing each point to its neighbors in feature space. It is important to note that the neighbor points are defined by the feature spaces of each layer, rather than their physical positions in 3D space. Unlike traditional CNNs that operate on regular grids in 2D image or 3D space, DGCNN dynamically updates the neighborhood structure in each layer, enabling it to capture complex 3D geometric properties through local feature aggregation and global max pooling. Our implementation used 1,024 points and 20 nearest neighbors, based on the publicly available code [26].

Point Transformer employs a self-attention mechanism inspired by transformer architectures in natural language processing and computer vision [27,28]. Each Point Transformer layer integrates information from neighboring points using attention weights based on spatial proximity and feature similarity. Transition Down layers progressively reduce the number of points in the input set, enabling hierarchical abstraction of point cloud representations. These layers select a representative subset of points via farthest point sampling [29] and aggregate local features from their k-nearest neighbors, allowing the network to capture increasingly coarse-grained geometric and semantic structures across layers. This downsampling process is analogous to hierarchical visual processing in the human visual system, where increasingly abstract and spatially compressed representations are formed at successive stages. In our implementation, we used $k = 16$ nearest neighbors following the original paper, and based our code on the publicly available implementation [30].

While both models operate on point cloud inputs, DGCNN emphasizes local geometric structure and information pooling from neighbors that share similar shape features, whereas Point Transformer combines hierarchical pooling and attention to capture context-dependent relations between a point and its neighbor points (defined in 3D space). To align with human experiments, we trained both models on the ModelNet40 dataset and extracted logits corresponding to the

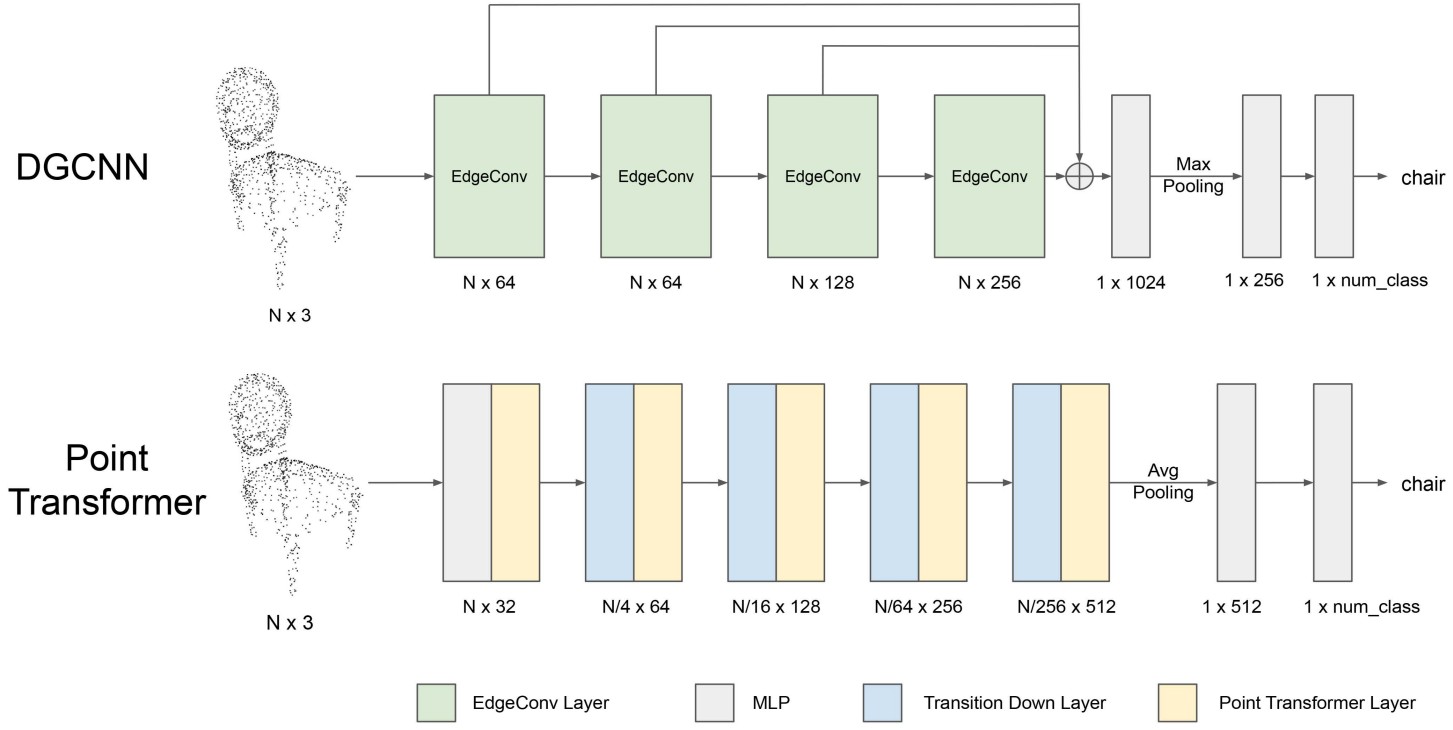

**Fig 2. The architectures of DGCNN (top) and Point Transformer (bottom).** *N*: number of points in each point cloud. MLP: Multi-layer perceptron, consisting of multiple fully connected layers. ⊕: concatenation.

**Table 1. Comparison and key differences between a convolution-based model (DGCNN) and a transformer-based model (Point Transformer).**

| DGCNN | Point Transformer |
|---|---|
| Feature embeddings of a reference point are modulated by adding the weighted feature differences from the neighbor points | Feature embeddings of a reference point are computed using feature embeddings of neighbor points weighted by similarity |
| Neighbor points are defined in feature space, changing from layer to layer | Neighbor points are defined based on distance in 3D space |
| No spatial resolution change | Downsampling: spatial resolution change from fine to coarse |
| No position (3D coordinates) encoding | Position encoding added to each layer |
| 1.81M parameters | 9.58M parameters |

ten object categories shown to participants, enabling direct comparisons of recognition performance. Both models were trained using the same data augmentation procedures, including random point dropout, random scaling, and random shifting of the input point clouds.

## Training process

To ensure a fair comparison among models, we adopted the data augmentation protocol implemented in the original Point Transformer repository [30] during training. Specifically, each point cloud was first subjected to random point dropout, where the dropout ratio, defined as the proportion of points to be removed, was uniformly sampled from the

range [0, 0.875]. The selected points were replaced by the coordinates of the first point in the cloud. Following this, each point cloud was uniformly scaled by a random factor drawn from the interval [0.8, 1.25] and randomly translated along the *x*, *y*, and *z* axes, with the translation magnitude for each axis independently sampled from a uniform distribution over [–0.1, 0.1].

For all models, training was conducted using the Adam optimizer with an initial learning rate of 0.001. Each model was trained for 200 epochs, and a step learning rate scheduler was employed to progressively reduce the learning rate during training, decreasing it by a factor of 0.3 every 50 epochs.

## Results

### General procedure

We used the same general procedure across all human experiments unless otherwise specified. Participants viewed a point cloud object in each trial. The stimulus was displayed for 3 seconds, after which ten buttons, each labeled with a different object name, appeared for selection. Their task was to select the object category that best matched the presented point cloud object. The ten object categories were airplane, bottle, bowl, chair, cup, lamp, person, piano, stool, and table. Each point cloud stimulus was presented as a GIF rotating 10 degrees per frame around the vertical axis. The GIF was displayed at 10 frames per second, completing a full 360-degree rotation in 3.6 seconds.

Participants first completed a practice trial showing a rotating point cloud of a plant. They had to select the correct object category before proceeding to the experimental trials. If participants selected a wrong object category during practice, then the practice trial was repeated, and a hint message was displayed below the ten category buttons, directing participants to select the "Plant" button. This practice trial aimed to familiarize participants with the point-cloud display and ensure they understood the recognition task. The object category, plant, in the practice trial was not included in the subsequent experimental trials, and the plant stimulus was not subjected to any experimental manipulations of point density, orientation, or deformation.

The experimental trials were similar to the practice trial, except that no feedback was provided. At the end of the experiment, demographic information was collected, and participants were presented with debriefing information about the study.

### Experiment 1: Point density and object orientation

In the first experiment, we investigated the recognition performance of both human participants and neural network models by manipulating: (1) point density in point cloud displays and (2) object orientation. By combining these conditions, we examined the robustness of human and model recognition across different levels of local detail and atypical viewpoints.

**Participants.** This study involved direct recruitment of human participants. Two groups of participants were recruited through the UCLA Subject Pool. For the upright condition, 56 participants were recruited (45 female, 11 male), with one participant excluded for reporting a lack of seriousness, resulting in a final sample of 55 participants (mean age = 19.8, SD = 1.4). For the inverted condition, an additional 47 participants were recruited (40 female, 7 male), with a mean age of 20.4 years (SD = 1.5). The average completion time for both conditions was approximately 10 minutes, with a slight variance between groups.

**Stimuli.** The stimuli were selected from the test set of the ModelNet40 dataset to ensure a fair comparison between human participants and deep neural network (DNN) models. We selected 7 object instances from each of the ten categories. Hence, the experiment included 70 different 3D object shapes. Each object stimulus was transformed by varying 1) point density: each point cloud was randomly downsampled to seven proportions (20%, 30%, 40%, 50%, 60%, 80%, and 100% of 1024 points). 2) object orientation: upright vs. inverted. Therefore, we generated 7 objects x 7 point densities x 10 categories = 490 stimuli for the upright condition and 490 stimuli for the inverted condition. Example stimuli with different densities and orientations are shown in Fig 3. Each point cloud stimulus was presented as a GIF rotating 10

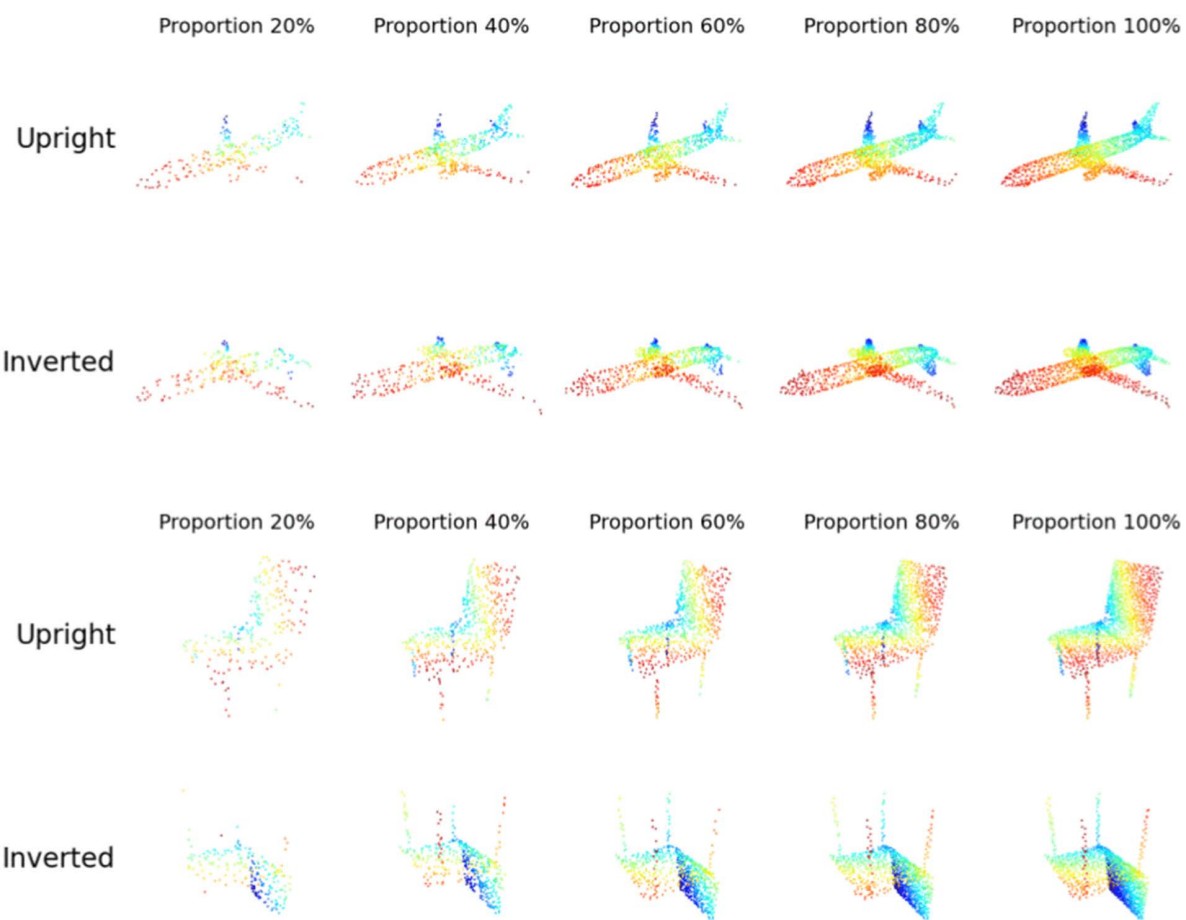

**Fig 3. Point cloud stimuli used in Experiment 1 by varying dot density and object orientation.** Colors represent depth, with red indicating proximity and blue indicating distance. In the experiments, the stimuli were presented as black points with rotation in depth, viewed from a horizontal viewpoint.

degrees per frame around the vertical axis. The GIF was displayed at 10 frames per second, completing a full 360-degree rotation in 3.6 seconds.

In the experiment, the participants were randomly assigned to either the upright or inverted conditions. Each participant viewed one object exemplar only once, with a random permutation of dot density. In other words, each participant viewed all seven objects from one category, and each object instance was displayed only once with randomly assigned point density. This resulted in a total of 70 trials per participant. The trials were randomized for each participant. For the models, we used all combinations, 490 objects x 2 conditions = 980 stimuli, for testing.

We adopted a between-subject design for orientation to avoid presenting the same objects in both upright and inverted positions, which could introduce familiarity effects and substantially increase the number of trials. In contrast, point density was varied within subjects to efficiently assess recognition robustness to visual sparsity while keeping the experiment length manageable. Each object exemplar was presented only once, and different exemplars from the same category were used across density levels, minimizing potential bias from prior exposure. We recognize that an alternative design—assigning distinct sets of objects to upright and inverted conditions—could also control for cross-condition familiarity. However, we prioritized a between-subject design for orientation to ensure clear interpretability of orientation effects and maintain a practical experiment duration.

**Results.** The results of the experiment are presented in Fig 4. Despite the sparse information provided in point cloud displays, human participants consistently demonstrated high accuracy across all levels of point density. Their performance ranged from 86.2% to 95.3%, with only a slight decline as the point density decreased. For instance, when the number of points was reduced to 20% of the original points, the mean accuracy was 86.4% (CI = [83.5%, 89.2%]), and at 30%, the mean accuracy was 88.9% (CI = [86.3%, 91.5%]). This result suggests that humans are highly resilient to reduced point density, maintaining reliable recognition performance even with sparse displays of 3D objects.

A mixed-design ANOVA was conducted with point density as a within-subject factor and orientation (upright vs. inverted) as a between-subject factor, with mean recognition accuracy as the dependent variable. The analysis revealed a significant main effect of orientation, $F(1, 100) = 25.58$, $p < .001$, indicating that participants in the upright condition achieved higher accuracy than those in the inverted condition. There was also a significant main effect of point density, $F(6, 600) = 25.54$, $p < .001$, showing that recognition performance varied across density levels, with lower accuracy for sparser point clouds. Moreover, the interaction between orientation and point density was significant, $F(6, 600) = 4.89$, $p < .001$, suggesting that the decline in recognition accuracy at lower point densities was more pronounced for inverted objects. These results indicate that both orientation and point density strongly influence recognition performance, and that orientation modulates the effect of density on visual recognition.

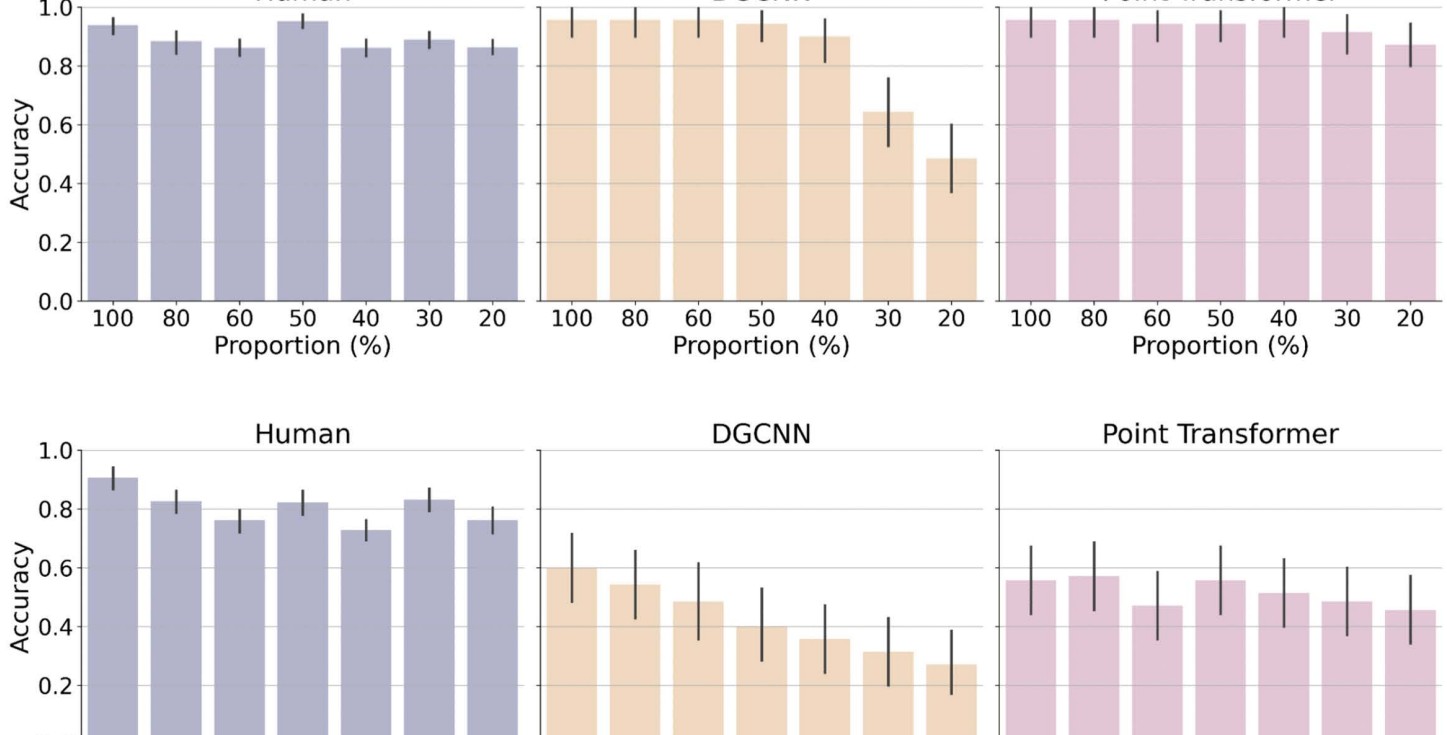

**Fig 4. Accuracy of human participants and models as a function of point density for upright point clouds (top) and inverted point clouds (bottom).** Error bars represent 95% confidence intervals around the mean accuracy, estimated across participants for human performance and across stimuli for model performance at each proportion level.

The performance of the DGCNN model, however, was markedly affected by the reduction in point density. While the model's accuracy approached human performance at high point densities (above 30%), it declined sharply at lower densities. Specifically, the accuracy dropped to 64.3% at 30% point density and 48.6% at 20%. In contrast, the Point Transformer model exhibited greater robustness to displays with low point density. For upright displays, its accuracy remained high across all levels of point density, ranging from 94.3% at 50% to 87.1% at 20%, showing similar performance robustness as humans.

To further examine error structure beyond overall accuracy, we compared the category-level confusion patterns of human participants with those produced by DGCNN and the Point Transformer (Fig 5). We quantified similarity by computing Pearson correlations between the off-diagonal elements of the confusion matrices (i.e., considering only misclassification patterns and excluding correct responses). The Point Transformer showed a strong correspondence with human error patterns ($r = 0.748$, $p < 0.001$), whereas DGCNN exhibited a weaker, though still significant, correspondence ($r = 0.474$, $p < 0.001$). These results indicate that the Point Transformer more closely captures the structure of human category confusions than DGCNN.

In the inverted condition, human participants showed the inversion effect with lower accuracy than upright condition, but still significantly greater than the chance level. Furthermore, humans consistently outperformed the machine learning models for the inverted objects, highlighting their adaptability to changes in viewpoint. Unsurprisingly, the models performed significantly worse with inverted 3D objects, as these stimuli were absent from their training set. This demonstrates the general limitations of deep learning models' strong reliance on training data. It is worth noting, however, that the Point Transformer model performed better than the DGCNN model overall and also showed less reduction in the inverted condition, particularly at lower point densities.

## Experiment 2: Lego-like point clouds

In Experiment 2, we systematically deformed the local geometric features of point clouds while preserving their global shapes. This was achieved by converting point clouds into voxel grid displays, analogous to reducing the resolution of an image. A larger voxel size leads to lower spatial resolution in the point cloud and increased local deformation. The process involved generating a voxel grid from the point cloud, sampling points on the voxel surfaces, and normalizing the sampled points. The stimuli section below details the methodology and the corresponding implementation.

The idea of introducing Lego-like point clouds was inspired by the sawtooth images from Baker et al. (2018), where sawtooth edges were added to silhouette images to disrupt local contour features. In our 3D point cloud display, we

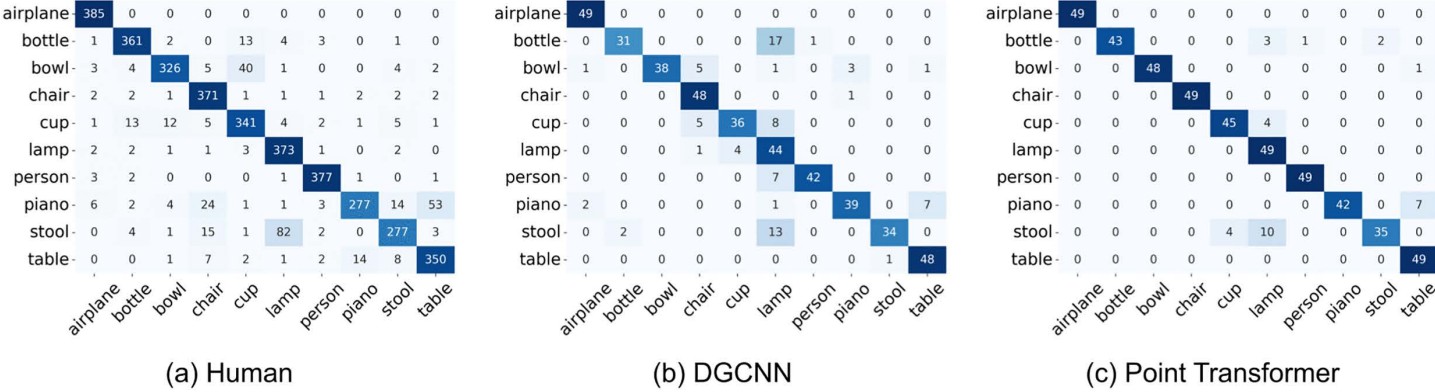

(a) Human (b) DGCNN (c) Point Transformer

**Fig 5. Confusion matrices for human responses, DGCNN predictions, and Point Transformer predictions in Experiment 1 upright condition.** Rows represent true labels and columns represent predicted labels.

similarly aimed to disrupt local 3D curvatures by converting point clouds into Lego-like displays while keeping the global shape almost unchanged.

**Participants.** This study involved direct recruitment of human participants. A total of 60 participants were recruited through the university's Subject Pool. The sample comprised 49 females, 9 males, 1 non-binary individual, and 1 participant who preferred not to disclose their gender. The mean age of the participants was 20.6 years (SD = 3.9). The average completion time for the experiment was 6.3 minutes (SD = 2.3).

**Stimuli.** We first converted each point cloud into a voxel grid representation using the Open3D library. A point cloud consists of discrete data points capturing an object's surface geometry, while voxels are small cubic units dividing 3D space into a regular grid, analogous to pixels in 2D images but extending into 3D space. To convert a point cloud into a voxel grid, we superimposed a voxel grid over the point cloud, determined voxel occupancy by checking which voxels contained points, and then created a structured, blocky representation of the object.

After obtaining the voxel grid display of the point clouds, we sampled points uniformly on the surface of this grid. These sampled points were aggregated to form a new point cloud constrained by the voxel grid. Varying the voxel size allowed us to sample the point cloud at different deformation resolutions. A visualization of the Lego-like point cloud stimuli across different voxel sizes is presented in Fig 6.

The resulting point cloud from the voxel sampling was then normalized to center it at the origin point, and scale it to fit within a unit sphere. This normalization did not affect the GIFs presented to human participants but was crucial for ensuring consistent spatial scale with the training set for machine learning models.

For the stimuli, we used the same 70 objects from 10 categories (that is, 7 object instances for each category) as in Experiment 1. Each object was sampled with four different voxel sizes: 0.01, 0.05, 0.1, and 0.2. Four instances of objects were randomly sampled from one category, and each instance was randomly assigned a voxel size, which made up a total of 40 stimuli per participant.

**Results.** The results of Experiment 2 are depicted in Fig 7. Human participants demonstrated relatively stable accuracy within a range of small local deformation, and then performance gradually decreased for more deformation with larger voxel size. Human performance remained almost the same when voxel size increased from 0.0 (mean accuracy 93.8%, $CI$ = [91.8%, 95.8%]) to 0.05 (mean accuracy 91.8%, $CI$ = [89.5%, 94.1%]). Then accuracy gradually reduced as the voxel size increased to 0.1 (mean accuracy 84.3%, $CI$ = [81.3%, 87.3%]) and 0.2 (mean accuracy 66.8%, $CI$ = [62.9%, 70.7%]). This suggests that human participants can effectively recognize objects even when local geometric features are deformed, provided the global shape remains intact.

Both models achieved performance levels comparable to human observers at small voxel sizes, suggesting their ability to generalize well with small local deformation. However, as voxel size increased, performance began to decline for both models. Notably, the Point Transformer exhibited a gradual decline in accuracy that closely mirrored the human trend, suggesting that it captures a similar sensitivity to degradation in local 3D curvatures. In contrast, DGCNN maintained stable performance up to voxel size 0.1 but showed a sharp drop at 0.2, indicating a threshold-like effect leading to brittle performance. Specifically, DGCNN's accuracy dropped from 95.71% at voxel size 0.1 to 74.29% at voxel size 0.2.

We further analyzed the error patterns by visualizing the confusion matrices for human participants, the DGCNN model, and the Point Transformer model, as shown in Fig 8. To quantify the similarity between human and model error patterns, we computed Pearson correlations between the off-diagonal elements of the confusion matrices (i.e., excluding correct classifications). The Point Transformer exhibited a moderate and statistically significant correlation with human error patterns ($r$ = 0.431, $p$ < 0.001), indicating a stronger alignment with human perceptual judgments at the category level. In contrast, DGCNN showed a weaker but still significant correlation with human confusion ($r$ = 0.247, $p$ = 0.019), suggesting some overlap in error patterns but a less human-like profile overall.

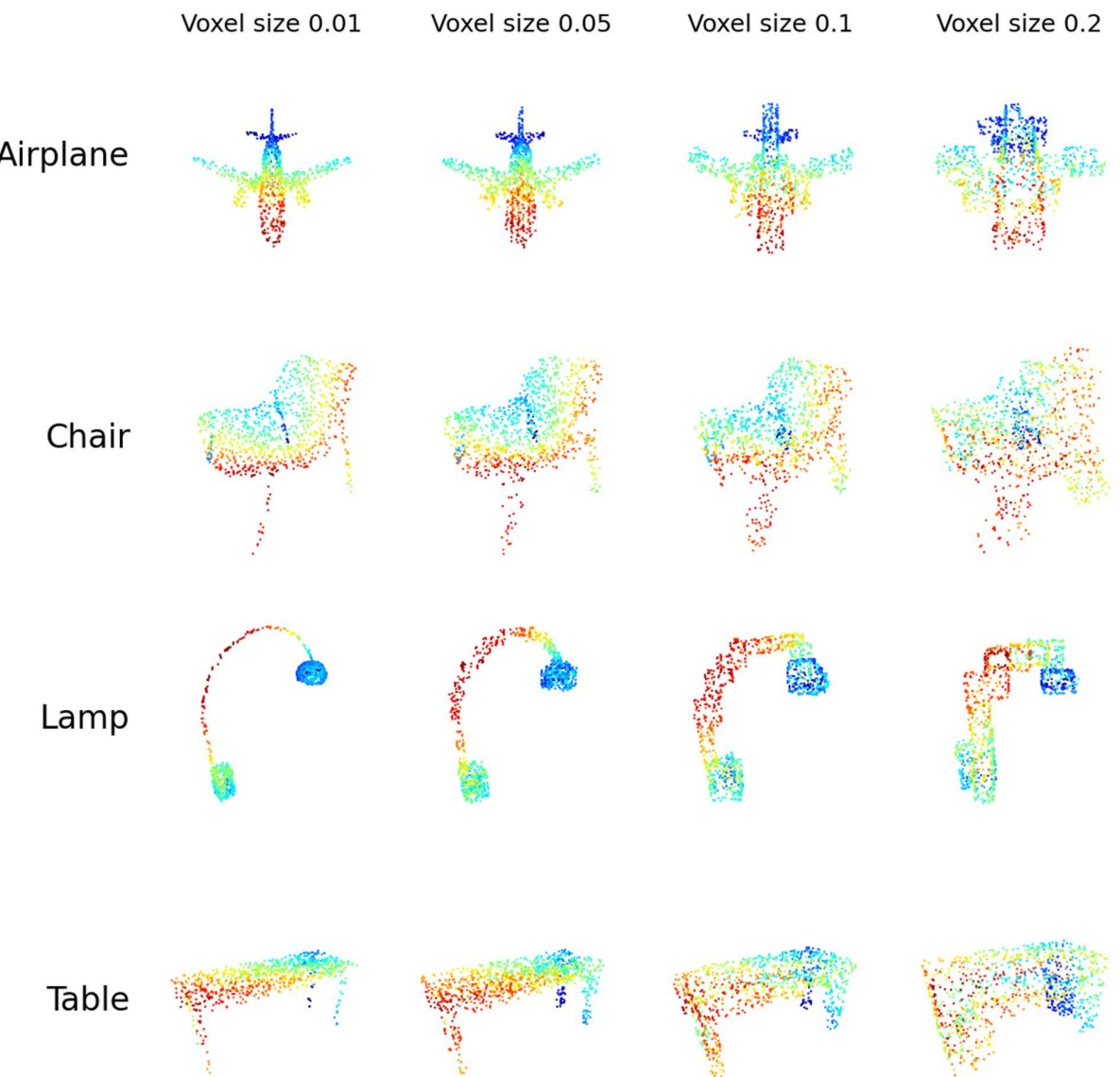

**Fig 6. Lego-like point cloud stimuli used in Experiment 2.** Points were uniformly sampled from voxel surfaces converted from point clouds, with varying voxel sizes determining the degree of local deformation. Higher voxel size indicates more local deformation.

## Experiment 3: part-scrambled point clouds

In Experiment 3, we discrupted the global shape properties while preserving the local geometric features of point clouds, in contrast to Experiment 2, which disrupted local curvatures but preserved global 3D shapes. Specifically, we used the part segmentation from the ShapeNet dataset [31] (ShapeNet-Part) which provides part labels for each point cloud, e.g., the wings, engines, body, and tail of an airplane. These parts were then randomly scrambled by placing them in different spatial locations to generate new point clouds that preserved local geometric features (parts) while rendering the global shape almost unrecognizable. If both human participants and deep learning models mainly relied on local geometric cues for object recognition, the part-scrambling manipulation would have no impact on recognition performance. Conversely, if recognition were based on global 3D shapes, this manipulation would lead to a substantial reduction in performance.

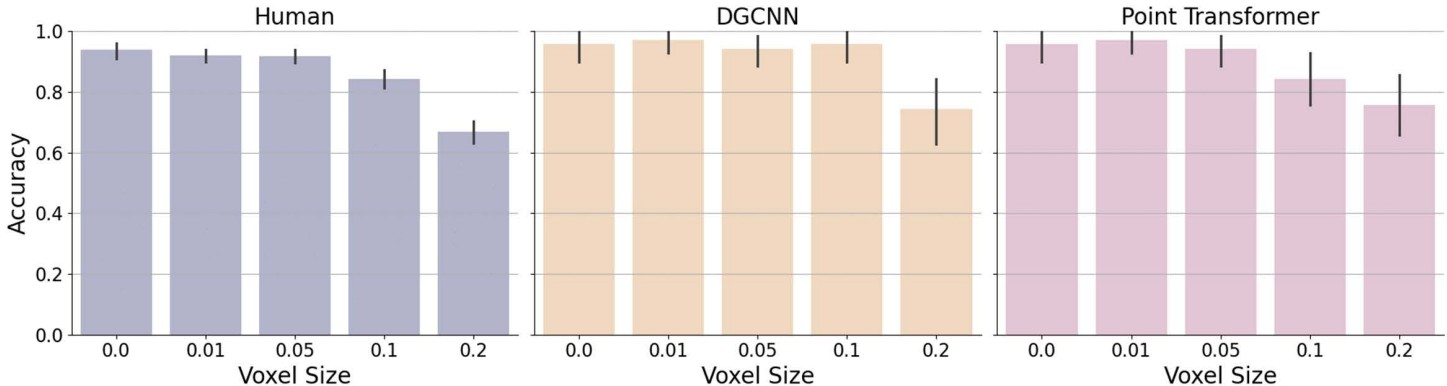

**Fig 7. Accuracy of human participants and models on Lego-like point clouds with varying voxel sizes.** Error bars represent 95% confidence intervals around the mean accuracy, estimated across participants for human performance and across stimuli for model performance at each voxel size level.

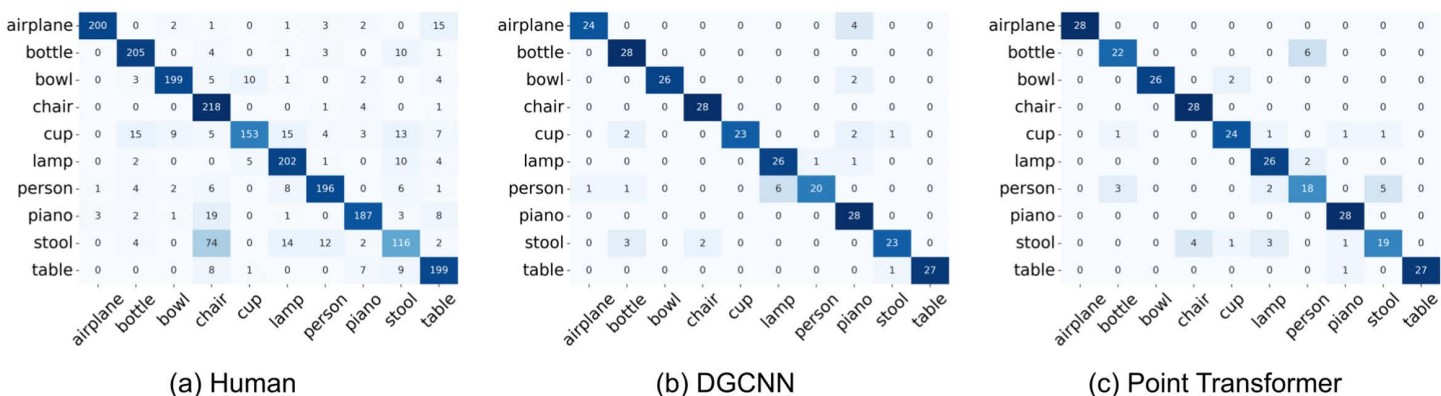

(a) Human          (b) DGCNN          (c) Point Transformer

**Fig 8. Confusion matrices for human responses, DGCNN predictions, and Point Transformer predictions in Experiment 2.** Rows represent true labels and columns represent predicted labels.

**Participants.** This study involved direct recruitment of human participants. A total of 89 participants were recruited through the university's Subject Pool, with 4 participants excluded for reporting a lack of seriousness. The final sample comprised 85 participants, including 69 females, 14 males, and 2 participants who preferred not to disclose their gender. The mean age of the participants was 20.2 years (SD = 1.74). The average completion time for the experiment was 20.56 minutes (SD = 9.73).

**Stimuli.** To generate part-scrambled point clouds, we used the part segmentation annotations provided by the ShapeNet-Part dataset [31], which is a subset of the larger ShapeNet repository and includes fine-grained part labels for 3D models in certain object categories. Since the object categories in ShapeNet-Part differed from those in the ModelNet40 dataset used to train our models, we focused on five overlapping categories: airplane, car, chair, lamp, and table.

For each object, we started with the part segments provided by the ShapeNet-Part dataset. If any individual part segment occupied more than 30% of the total point cloud (e.g., the body of a car), we further subdivided that segment using spectral clustering algorithms. Specifically, we applied the Spectral Clustering algorithm with 3 clusters and

nearest-neighbor affinity to break down large segments into finer parts. The number of part segments was in the range of 3–12 for the experimental stimuli.

After segmentation, each part was subjected to random transformations to disrupt the global configuration while preserving local geometric structure. One part was randomly designated as the anchor and centered at the origin by subtracting its centroid; this anchor remained fixed. Each remaining part was (1) centered by subtracting its own centroid, and (2) displaced by adding a random 3D offset sampled uniformly from a cube of side length $2 \times r$ (i.e., in the range $[-r, r]^3$, with $r = 0.6$). Finally, the entire scrambled point cloud was globally normalized to fit within a unit sphere to match the input scale used during model training.

Given the vast space of possible part permutations, not all scrambled variants are informative or recognizable by models. To ensure experimental control, we selected part-scrambled point clouds for which each model (DGCNN or Point Transformer) produced the same classification label as for the original, unscrambled point cloud.

For each model, we randomly selected 10 original point clouds from each of the five object categories, resulting in 50 object instances per model. For each object instance, we generated one part-scrambled version, yielding 50 part-scrambled stimuli. This resulted in 100 stimuli per model (50 original + 50 scrambled). Across the two models, the total number of stimuli was 200 (see Fig 9 for examples).

**Results.** The results of Experiment 3 are presented in Fig 10. Humans performed near ceiling on intact point clouds (mean accuracy: 97.5%, 95% CI = [97.2%, 97.9%]), with consistently high accuracy across all five object categories. However, recognition accuracy dropped sharply when global shape was disrupted by part scrambling. For stimuli scrambled according to DGCNN predictions, accuracy fell to 70.0% (CI = [68.6%, 71.4%]). For stimuli scrambled based on Point Transformer predictions, accuracy was slightly higher at 75.2% (CI = [73.9%, 76.5%]).

At the category level, the largest decrease was observed for *tables*, which dropped from 95.4% in the intact condition to 44.7% (DGCNN-scrambled) and 57.6% (Point Transformer–scrambled). Significant declines were also observed for the *car* and *chair* categories, while *airplanes* showed relatively smaller performance reduction (approximately 11%) in part scrambling conditions.

We next assessed model robustness by testing each model on scrambled stimuli generated from the other model's predictions. Fig 10 middle and bottom panels show recognition performance from DGCNN and Point Transformer, respectively. DGCNN achieved an average accuracy of 74% on Point Transformer–scrambled inputs, whereas Point Transformer performed notably worse, achieving only 64% on DGCNN-scrambled inputs. This asymmetry suggests that Point Transformer relies more heavily on global spatial structure for object classification and is more sensitive to disruptions in overall shape configuration. In contrast, DGCNN appears more robust to such disruptions, likely due to its bias toward local geometric features.

To further assess the alignment between model and human behavior, we computed Pearson correlations between the human and model confusion matrices (excluding the diagonal). The correlation between Point Transformer and human confusion was modestly positive ($r = 0.2417$, $p = 0.305$), while the correlation for DGCNN was near zero ($r = -0.0135$, $p = 0.955$). These results indicate limited correspondence between model and human error patterns, though Point Transformer showed slightly better alignment.

Finally, we compared the accuracy patterns of the DGCNN and Point Transformer to human accuracy patterns across all experimental conditions in the three experiments. By pooling performance data across all tested conditions (varying point density, object orientation, local deformation, and part scrambling), we computed Pearson correlations between each model's accuracies and human responses. We found that the Point Transformer model showed a higher correlation ($r = 0.618$, $p < 0.001$) with human performance than did the DGCNN ($r = 0.534$, $p = 0.019$). For a comprehensive analysis, we refer readers to section "Model and human performance correlation."

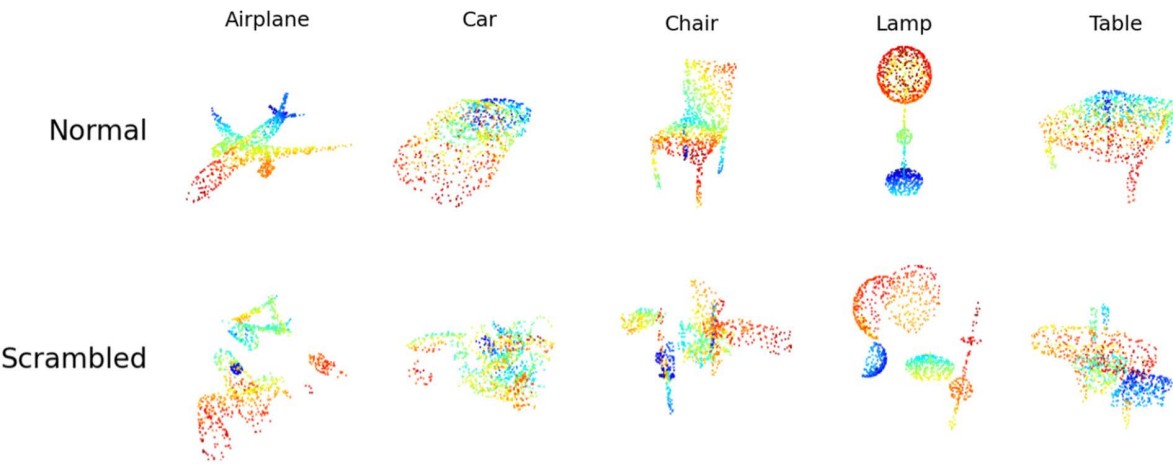

(a) Scrambled point clouds generated based on DGCNN predictions.

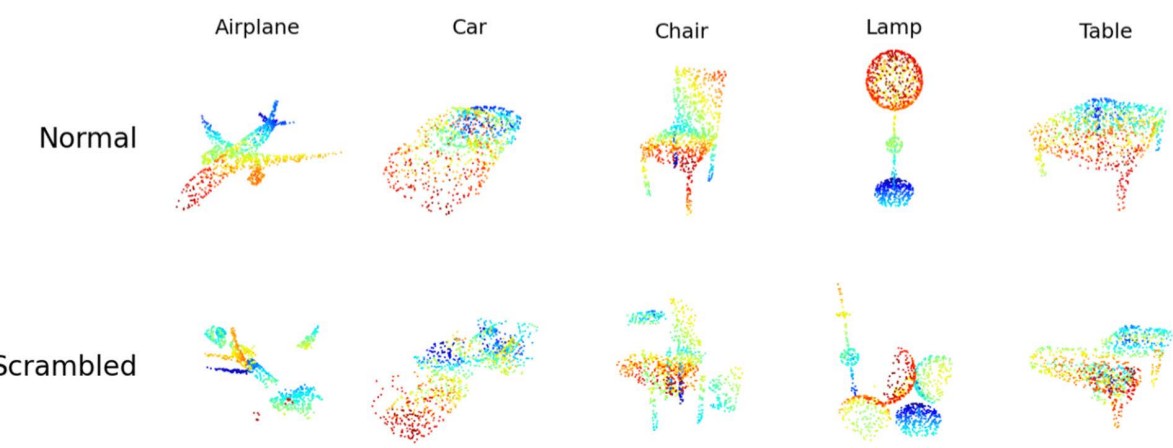

(b) Scrambled point clouds generated based on Point Transformer predictions.

**Fig 9. Scrambled stimuli used in Experiment 3.** Each row shows five objects (airplane, car, chair, lamp, and table) in either the original (top row) or scrambled (bottom row) condition. Scrambling was performed separately for DGCNN and Point Transformer models while preserving part identity.

## Ablation simulation results using variants of point transformer

To identify which computational mechanism in the Point Transformer model contribute to its global shape sensitivity, we systematically evaluated three primary differences between the Point Transformer and the DGCNN involving: (1) the Attention mechanism, (2) Position Encoding, and (3) the Downsampling operation supporting hierarchical pooling and abstraction of 3D shapes. The attention mechanism used vector self-attention, an extension of standard self-attention that produces a learned attention vector rather than a scalar for each neighbor, to differentially weigh the influence of each local neighbor based on both their content similarity and spatial configuration. Points with high similarity received higher attention weights during this integration process. The position encoding component enables 3D point coordinates to play a role in computing similarity in every layer. Specifically, similarity is computed by integrating both the similarity of feature embeddings and the spatial proximity between

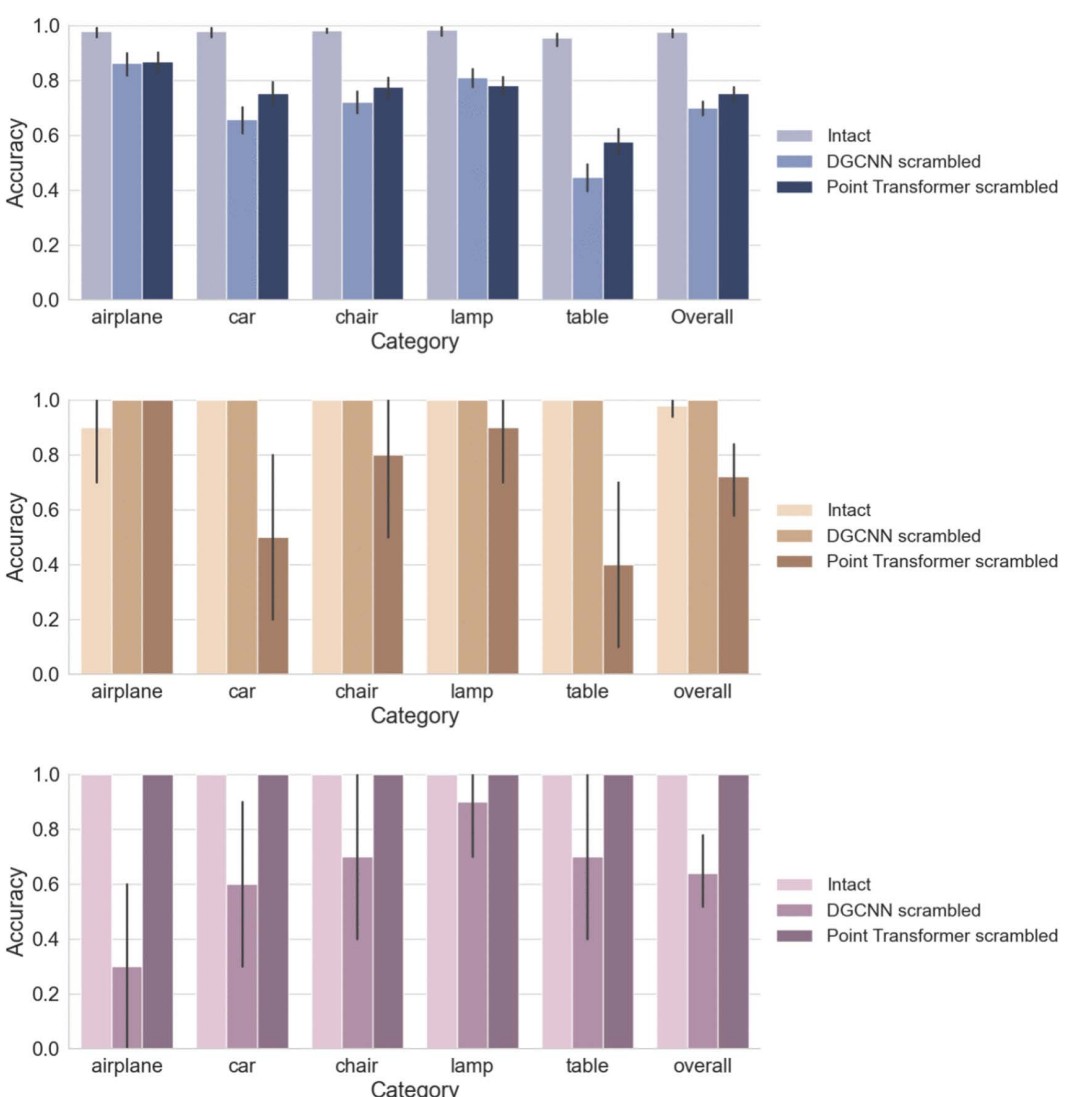

**Fig 10. Recognition performance of humans and models under scrambling manipulations.** Top panel, Human classification accuracy for intact and scrambled point clouds. Middle panel, DGCNN model accuracy for intact and scrambled point clouds. Bottom panel, Point Transformer model accuracy. Error bars represent 95% confidence intervals around the mean accuracy, estimated across participants for human performance and across stimuli for model performance.

points in the 3D space. Finally, the downsampling operation, implemented via Transition Down modules, progressively reduces the number of points across layers at a fixed ratio [$N$, $N/4$, $N/16$, $N/64$, $N/256$] for $N$ input points, thereby pooling local information into increasingly abstract global representations [18]. The downsampling mechanism effectuates an inductive bias of pooling information from local to global through abstraction, mimicking hierarchical visual processing in biological systems.

By selectively removing each component from the Point Transformer model, we trained multiple variants of the Point Transformer and then assessed their performance on the same point cloud stimuli used in human experiments. In these ablation simulations, the model variants were trained and tested using the same procedure as the original Point Transformer model except removing each of the components of interest.

Fig 11 illustrates the accuracy of different Point Transformer variants as a function of point density, stimuli used in Experiment 1. The original Point Transformer model, the "Original" variant in the plot, achieves the highest performance across all proportions. The "NoAttn" and "NoPE" variants, which remove the attention mechanism and the position encoding mechanism respectively, experiences a mild accuracy drop compared to the original. This suggests that while attention and position encoding are beneficial, the model maintains substantial effectiveness in their absence.

In contrast, the variant lacking the downsampling mechanism ("NoDS") demonstrated performance comparable to the original model at higher point densities but exhibited significant accuracy declines at lower point densities. This result underscores the critical role of the downsampling mechanism in generalizing across varying degrees of point density. Downsampling mechanisms facilitates hierarchical processing within the model, enabling global integration of local shape information and reducing dependence on local features.

Interestingly, this effect mirrors the hierarchical processing found in the human brain's ventral visual pathway. This is evidenced by the increasing receptive field size of neurons as one moves from low-level to high-level visual areas, allowing for the processing of information from local to global scales. Similarly, Downsampling in Point Transformer enforces global shape sensitivity by acquiring representations for fewer points sampled from the 3D shape in later layers. By progressively reducing the number of points covering an entire object, downsampling allows the model to integrate local features into a coherent global representation, making it more robust to various data transformations.

## Intervention simulation: enhancing DGCNN with downsampling

Given the critical role identified for the downsampling mechanism for fostering a global shape sensitivity in the Point Transformer model, we next examined whether adding this computational mechanism into the standard DGCNN architecture could enhance its performance. We added the Transition Down layer from the Point Transformer after each Edge-Conv layer of the DGCNN, matching the structural depth of the original Point Transformer.

As demonstrated in Fig 12a, incorporating the Transition Down layer significantly improved the DGCNN's robustness to point density. The original DGCNN exhibited significant performance degradation at point densities less than 40%. In contrast, the modified DGCNN with the Transition Down layer ("DGCNN+DS") increased the accuracy for the lowest point

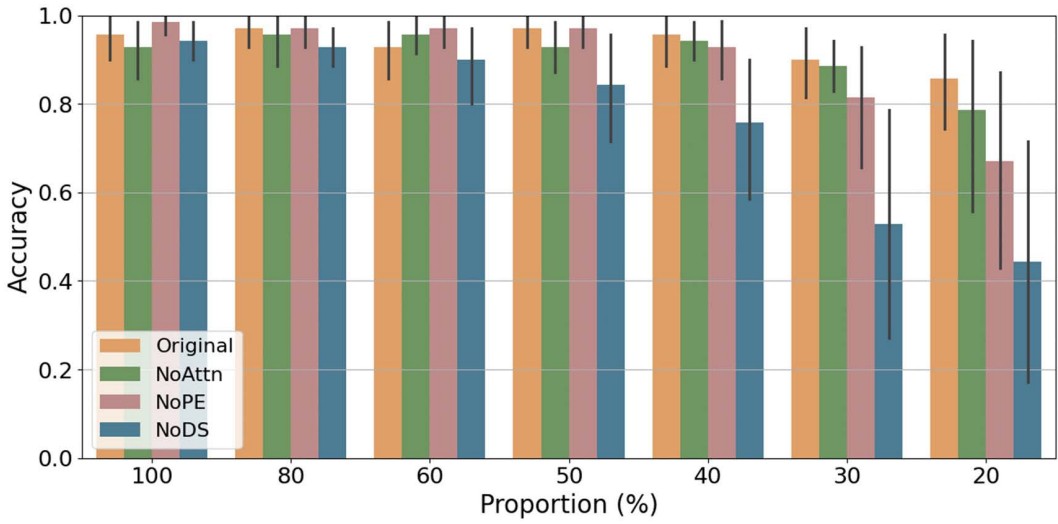

**Fig 11. Accuracy of variants of Point Transformers as a function of point density.** Original = the original Point Transformer, NoAttn = Remove attention, NoPE = Remove position encoding, NoDs = Remove downsampling.

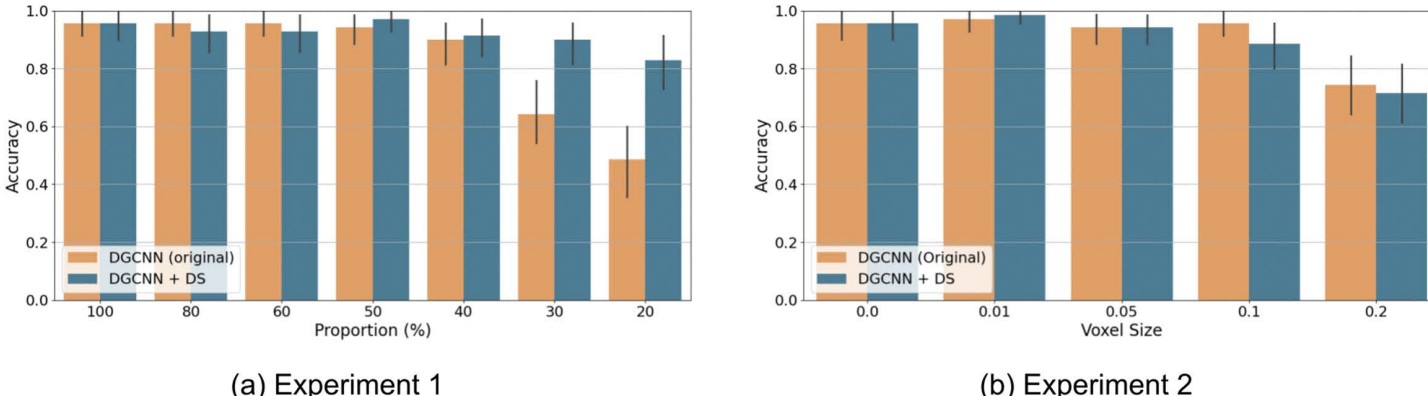

**Fig 12. Accuracy of variants of DGCNN models as a function of varying point density.** DGCNN+DS = DGCNN model with an additional Downsampling layer after each EdgeConv layer.

density conditions. For example, performance increased from 0.486 to 0.829 for the 20% point density, aligning closely with both human performance and the original Point Transformer. Results in Fig 12b suggest that DGCNN+DS displays a progressive reduction in accuracy as voxel size increases, consistent with the performance trend observed for the Point Transformer and humans. The original DGCNN exhibits an abrupt decline at voxel size 0.2 as discussed in Experiment 2. Quantitative correlation analyses are reported in the following section.

The addition of the Transition Down layer effectively compels the model to construct more abstract, hierarchical representations of 3D objects, shifting its focus towards global shape characteristics rather than relying excessively on local features. Crucially, this demonstrates that Downsampling can effectively bridge the gap between seemingly distinct model architectures, transformer-based and convolution-based, highlighting a shared computational mechanism toward robust shape recognition. Within the scope of point cloud recognition, where inputs are irregular, unordered surface samples and density perturbations directly alter local neighborhood statistics, hierarchical aggregation enabled by downsampling emerges as a key contributor to stable and human-like shape recognition. The consistent benefit of downsampling across both convolution-based and transformer-based point-cloud models further suggests that hierarchical abstraction provides a general and effective strategy for improving robustness to sampling variability in 3D object recognition.

## Comparison of model and human performance

We compared the accuracy patterns of the DGCNN and Point Transformer, along with their respective variants, DGCNN+DS and Point Transformer noDS, to human accuracy patterns across all experimental conditions in the three experiments. By pooling performance data across all tested conditions, we computed Pearson correlations between each model's accuracies and human responses.

As shown in Fig 13, both model variants that incorporate the Downsampling mechanism (DGCNN+DS and Point Transformer) exhibited substantially higher correlations with human performance than their non-downsampling counterparts (DGCNN and Point Transformer noDS). Specifically, the correlation between human performance and DGCNN+DS was $r = 0.684(p < 0.001)$, and for the original Point Transformer it was $r = 0.618(p = 0.005)$, both significantly higher than the correlation with the original DGCNN ($r = 0.534, p = 0.019$) and Point Transformer noDS ($r = 0.544, p = 0.016$). These differences were statistically supported by comparisons such as DGCNN vs. DGCNN+DS ($z = -13.239, p < 0.001$) and DGCNN vs. Point Transformer ($z = -6.123, p < 0.001$), indicating that the introduction of downsampling yields not only improved model robustness but also closer alignment to human accuracy profiles.

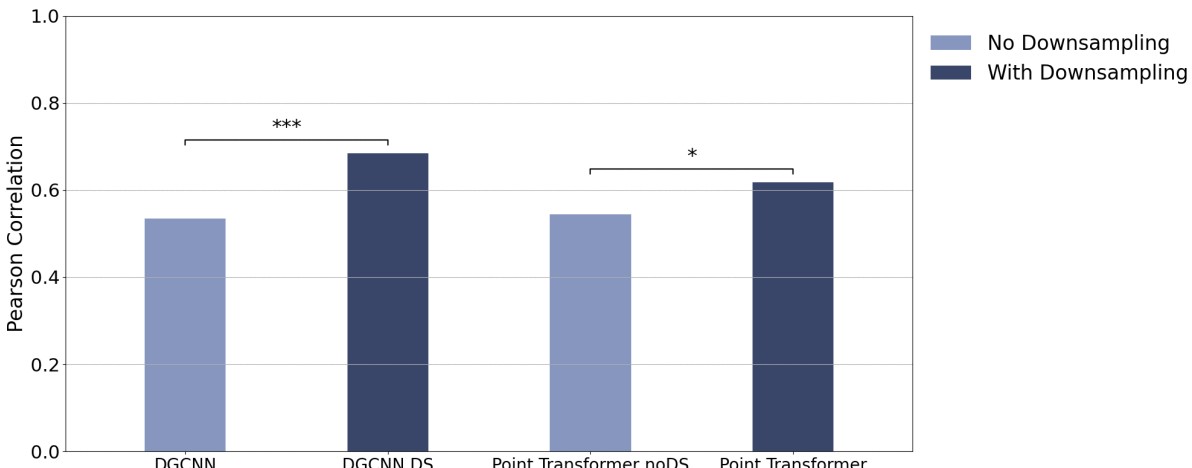

**Fig 13. Correlation between model and human accuracy patterns across all experiment conditions.** Models with downsampling show significantly stronger alignment with human.

In addition to the two primary architectures, we evaluated three further point-cloud classification models, PointNet++ [29], PointNeXt [32], and the Point Cloud Transformer (PCT) [33], all of which incorporate a downsampling mechanism and are trained under the same condition. These models were tested on the same stimulus set from the three experiments, and their accuracy patterns were compared with human performance using the same correlation-based analysis. All three models exhibited moderate-to-strong alignment with human accuracy patterns: PointNet++ yielded a correlation of $r = 0.632$ ($p = 0.003$), PointNeXt yielded $r = 0.636$ ($p = 0.003$), and PCT yielded $r = 0.632$ ($p = 0.003$).

This finding provides converging evidence that hierarchical processing implemented via a downsampling mechanism plays a central role in eliciting human-like recognition of 3D shapes, and the enhanced correspondence across structurally distinct architectures underscores its broad applicability. Accordingly, downsampling-based models align more closely with human performance than non-downsampling variants such as DGCNN and Point Transformer noDS.

## Conclusion and discussions

Across three behavioral experiments and a series of modeling analyses, we demonstrated that humans exhibit robustness in recognizing 3D objects, even under challenging conditions such as sparse input, inversion, or disrupted local geometry. This is the first study to employ point cloud stimuli across a broad range of object categories and a large number of object instances. While current deep learning models—DGCNN and Point Transformer—achieved comparable accuracy to humans under standard conditions, their robustness to deformation under untrained conditions varied substantially. Our results build on and extend previous research, confirming that deep learning models of object recognition are adept at extracting local features (both image and geometric), but struggle to capture global shapes in both 2D and 3D.

By testing different deep learning models, however, we also demonstrated that their sensitivity to global shapes varies significantly. In particular, the Point Transformer model consistently mirrored human performance qualitatively, showing gradual declines when reducing point density and introducing local geometric deformation. In contrast, the DGCNN model was more brittle, with sharp drops in performance in these untrained conditions. These findings indicate that some deep learning models relying predominantly on local geometric features, such as DGCNN, lack the holistic processing strategies characteristic of human perception.

To identify the core computational mechanism in support of global shape sensitivity, our ablation studies revealed that the Downsampling mechanism in Point Transformer contributes the most to robust 3D object recognition. We confirmed

this by introducing this mechanism into the DGCNN model, which significantly improved its performance and alignment with human responses. This finding challenges the common assumption that attention mechanisms are the main contributors to generalization in transformer-based models. Instead, we propose that abstraction through downsampling, paralleling hierarchical visual processes in biological systems, plays a more critical role in producing robust 3D shape recognition.

Our results highlight the importance of global shape representations for human-like 3D object recognition. We found that integrating cognitively-inspired mechanisms, such as abstraction through hierarchical processing, into deep learning models can improve their generalization and robustness. However, the global shape sensitivity in human vision may arise from multiple routes. The visual system provides the brain with a wealth of information about the physical properties of objects.

Beyond simply recognizing categories, accurately estimating 3D shapes is crucial for action, motor planning, and reasoning. Global shape sensitivity, both 2D and 3D, arises early in human development, and evidence from both cross-species and human infant studies suggest that it builds on evolutionarily endowed mechanisms [34]. In addition to these early-developing perceptual capacities, active vision further enhances global shape sensitivity through interactions between perception and action following the onset of goal-directed reaching and locomotion. Such interactions can be viewed as a form of multisensory data augmentation for acquiring global shape representation of 3D objects. Although point cloud displays are novel for human observers and differ substantially from how 3D objects are typically encountered in the natural environment, humans nonetheless perform robustly on these stimuli. By contrast, data augmentation in deep learning models remains relatively limited in scope. In the present work, we followed the standard model implementation by augmenting training data through variations in point density and viewing angle, which improved model performance but is unlikely to fully account for human-level robustness. We therefore suggest that humans rely on richer forms of data augmentation, such as active exploration and manipulation of objects, to support the acquisition of 3D shape representations. Because current machine vision models lack both innate, specialized perceptual mechanisms and also embodied cognition, their ability to acquire global shape representations is limited.

By systematically manipulating local geometric features and global shape structure, our paradigm provides a well-controlled empirical test using point clouds for examining both the development of shape perception and its impairments in aging and clinical populations. Furthermore, our study demonstrates that introducing an inductive bias through architectural design is a promising way to learn 3D shapes of objects. However, other approaches could further enhance this ability. Future work could explore other biologically inspired architectures to test whether similar global shape sensitivity emerges through different computational principles. For example, spiking neural networks, which model temporal coding and neural dynamics more faithfully to biological neurons, could offer an alternative and more efficient framework than attention-based transformer architectures for examining the origins of global shape bias [35,36]. Comparing such models with hierarchical architectures could clarify whether human-like robustness arises primarily from structural organization or from the intrinsic dynamics of neural computation. In parallel, future studies could incorporate recurrent feedback of biological vision systems [37,38], or diversify training data with egocentric, action-based experiences that expose models to objects from multiple viewpoints and contexts of interaction [39–41]. Exploring these directions would be a significant step toward developing more biologically grounded and capable, agentic AI systems.

## Acknowledgments

We thank Daniel Tjan and Zhiqi Zhang for their invaluable assistance with data collection.

## Author contributions

**Conceptualization:** Shuhao Fu, Philip J. Kellman, Hongjing Lu.

**Data curation:** Shuhao Fu.

**Formal analysis:** Shuhao Fu.

**Funding acquisition:** Hongjing Lu.

**Investigation:** Shuhao Fu, Hongjing Lu.

**Methodology:** Shuhao Fu, Hongjing Lu.

**Supervision:** Philip J. Kellman, Hongjing Lu.

**Validation:** Shuhao Fu, Philip J. Kellman, Hongjing Lu.

**Visualization:** Shuhao Fu.

**Writing – original draft:** Shuhao Fu, Hongjing Lu.

**Writing – review & editing:** Shuhao Fu, Philip J. Kellman, Hongjing Lu.

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
