## [Decision Letter · Decision Letter 0]

20 Oct 2025

Hierarchical abstraction drives human-like 3-D shape processing in deep learning models

PLOS Computational Biology

Dear Dr. Fu,

Thank you for submitting your manuscript to PLOS Computational Biology. After careful consideration, we feel that it has merit but does not fully meet PLOS Computational Biology's publication criteria as it currently stands. Therefore, we invite you to submit a revised version of the manuscript that addresses the points raised during the review process.

Please submit your revised manuscript within 60 days Dec 20 2025 11:59PM. If you will need more time than this to complete your revisions, please reply to this message or contact the journal office at ploscompbiol@plos.org. Please include the following items when submitting your revised manuscript:

We look forward to receiving your revised manuscript.

Kind regards,

Jian Liu

Academic Editor

PLOS Computational Biology

Marieke van Vugt

Section Editor

PLOS Computational Biology

**Additional Editor Comments:**

This manuscript presents a comparative study of human and deep learning performance in 3D object recognition. To strengthen its contribution, the work would benefit from a more detailed explanation of the model, a more rigorous validation of the experimental tests, and a more systematic discussion of prior research to better frame its novel insights.

**Journal Requirements:**

3)Kindly revise your competing statement in the online submission form to align with the journal's style guidelines: 'The authors declare that there are no competing interests.'

**Reviewers' comments:**

Reviewer's Responses to Questions

**Comments to the Authors:**

Reviewer #1: Summary: This manuscript compare human and two types of DNN on recognizing object from 3d point cloud. The authors found that human show robustness to “drop points” “invert” “Legofy”; transformer-based DNN is moderately more robust than convolution-based DNN. The authors further identified the key component, down-sampling, that causes the robustness difference between the two models.

The logic of the manuscript is legit, though I have a few concerns on the detailed experimental design, and interpretations of the results. I would like to see the authors adding discussions (potentially figures) to the main text on the following points.

1. Effects of data augmentation

The model performance on the density drop experiment can be closely related to how the model is trained, specifically how does the data augmentation drops the points. In the method section, the authors stated “random point dropout” is used, but details on the dropout ratio are not reported. If the dropout ratio matches the “drop proportion” in the experiment, the model could perform well. Then the question is, if the above is true, such that the performance depends on the training configs, how to interpret the result, such as “Point Transformer is more robust and more similar to human performance”. Is it that the model is more robust, or the training config makes it more robust? The authors should discuss the effects of training configs.

The model and training details in the method section are minimal. The authors should add details such as learning rate, epoch, model size, etc.

2. Questions on Experiment 3

In Experiment 3, are subjects informed that the parts are scrambled? If they are not (though it is close to the DNN setting; DNN doesn’t see any instruction), this would be a confusing task. If they are, then it is not a fair comparison to DNNs. The authors should discuss this point.

(minor)

In Fig 8. The examples given in a) and b) are different. Choosing the same instance but scrambling based on DGCNN and Point Transformers can be more informative.

A broader question is whether the labels are still meaningful after scrambling. The authors may get the inspiration from the known texture-shape bias in CNN and human perception. However, I believe part-whole bias is less natural to study compared to other biases. In studies on texture-shape bias (Geirhos, Robert, et al. 2018, should cite in the introduction when the authors talk about CNN uses less global information), the visual concepts they used have a stereotypical texture and shape (e.g. elephant vs cat); from either texture or shape, the subject should be able to identify the visual concept. For the same reason, the design of Baker 2019 et. al. is also valid. However, in the experiment 3 on part-whole, I believe parts only do not contain sufficient information to identify the object, especially given that classes like chair and table are similar. The parts can be just simple simplices.

3. Overclaims of the ablation and intervention experiments

Line 275-279, the authors claim Point Transformer relies more on global structure, because its acc on scrambled inputs is 64% compared to 74% in DGCNN. However, Fig 9. shows a large variability across categories. It actually looks like Point Tansformer is better than DGCNN on 3 over 6 categories. The low relative accuracy is likely due to DGCNN has a surprising perfect accuracy on airplane, which is likely due to the Point Transformer-based scramble of airplane preserving the global shape. This is a huge confounder for making the above conclusion. The experiment design would be cleaner if using the same scrambled inputs on both networks.

Line 332-334, the authors claim “Downsampling mechanisms facilitate hierarchical processing within the model, enabling global integration of local shape information and reducing dependence on local features”. The evidence from the point density drop experiment does not directly support this claim. The experiment drops points randomly across the whole point cloud. It is unclear whether it disrupts more local features than global shape features. The Experiments 2 and 3 would be more direct support. Similarly, the “Intervention simulation” section is tested on density drop, but should also test other experiments. The authors indeed did those experiments and use the results to compute the correlation between human performance in Fig 12. The results on each task should be reported to show which task differs the most with and without downsampling.

Minor issues:

1. Line 363-366, “While it is commonly assumed that the attention mechanism in transformer-based architectures primarily enables better generalization, our results clearly demonstrate that it is actually the Downsampling mechanism driving this effect.” This claim should be more specific than general. In general, most of the transformer-based architectures do not have downsampling, but convolution-based architectures have downsampling. And the authors should discuss why this 3d point cloud case is different from other cases on different modalities, such as 2d images, time sequence, etc.

2. Fig 1. Missing the meaning of color. I assume they indicate depth.

3. Fig 2. Missing definition of N. I assume the number of points.

Reviewer #2: Summary

This manuscript investigates whether deep learning models develop human-like 3D shape representations for object recognition. The authors conducted three well-controlled human behavioral experiments, systematically manipulating point density and object orientation, local geometric structure, and part configuration. Human participants consistently outperformed the models under all experimental conditions. Among the evaluated architectures, the Point Transformer provided a closer approximation to human performance than the convolution-based DGCNN. Through ablation and intervention analyses, the study further identifies that the main advantage of the transformer model arises from the downsampling mechanism, which supports hierarchical abstraction of 3D shapes.

Strengths

1. The integration of human behavioral experiments with computational modeling provides strong empirical grounding and enhances the scientific credibility of the conclusions.

2. The three complementary experiments systematically examine local versus global shape processing, offering a multi-perspective and rigorous comparison between humans and models.

3. The inclusion of ablation studies and architectural interventions effectively demonstrates that hierarchical downsampling is the key factor driving human-like robustness in 3D recognition.

4. All datasets, stimuli, and code are publicly available, meeting high standards for open science and enabling replication and further research in this field.

Weaknesses

1. Each experiment recruited only a few dozen participants with an average age around 20 years, likely undergraduate students. This narrow demographic range limits generalizability. Including more diverse participants or providing justification for this selection would strengthen the validity of the findings.

2. Human participants possess rich visual experience, while the models were trained solely on the ModelNet40 dataset. Future work should test model generalization on more diverse or self-supervised 3D datasets to better approximate human learning conditions.

3. Only two neural architectures were compared. Incorporating additional representative models, such as PointNet++ or recent hybrid graph-transformer architectures, would enhance the robustness and generality of the conclusions.

4. It's not just the network architecture. I think brain-like neural networks, such as spiking neural networks, are also worth trying, so that we can know whether the results are related to the neuron model or the network structure.

5. The comparisons rely mainly on classification accuracy. It would be valuable to include representational similarity analysis (RSA), confusion matrix correlation, or embedding-space visualization to deepen the understanding of model–human correspondence.

6. While detailed, the manuscript could benefit from greater focus. Streamlining sections with redundant experimental details and emphasizing key theoretical contributions would improve readability and highlight the central message.

7. Although the discussion draws analogies between downsampling and hierarchical visual processing in the human brain, the paper would benefit from citing or connecting more directly to neuroscientific evidence supporting hierarchical abstraction in biological vision.

Reviewer #3: This manuscript investigates the correspondence between human and deep learning model performance in recognizing 3D objects from point-cloud representations. Overall, the study is technically well executed and the topic is interesting. However, I have significant concerns regarding the conceptual framing and scientific contribution, which lead me to recommend rejection dicision.

Major issues:

1. Although the work is methodologically thorough and the question is potentially meaningful, the central contribution lies in analyzing model performance differences rather than offering insights into biological mechanisms. The study primarily performs *model–human comparisons* to infer internal properties of deep learning models, but it does not provide new understanding about the neural or computational principles of biological vision. As such, the manuscript reads more as a machine learning study on architectural design than a biological modeling paper, which falls outside the core scope of *PLOS Computational Biology*.

2. The title emphasizes “hierarchical abstraction drives…” as if the study provides direct evidence or theoretical grounding for hierarchical abstraction in biological visual systems. For me, it is very hard to link “hierarchical abstraction” and their experiments & results. The title and framing should therefore be reconsidered.

Other major concerns:

1. Introduction flow - The Introduction reads as a sequence of descriptive paragraphs rather than a clear argument that motivates the key scientific question.

2. The authors missed a lot of method details. e.g., It is unclear whether the stimuli in the practice trials differed in design or difficulty from those in the main task. The rationale for this distinction should be clarified.

3. We need to see more statistical reports. e.g., Figure 4 and related results lack statistical details. Given the multiple conditions (point density, orientation), the authors should report relevant statistics (e.g., ANOVA main effects and interactions) and show *p* and *t* values directly on the figure. For the upright vs. inverted comparison, a proper test of orientation effects is warranted.

4. The manuscript does not explain how model error bars were derived (across random seeds? test batches?). This is essential for interpreting Figures 4–9 and assessing model variability.

5. In Fig 9, the DGCNN model shows higher accuracy for “scrambled” than for intact point clouds, which is counterintuitive and undermines the claim of human-like correspondence. The authors should address why this occurred and what it implies for their interpretation.

6. The authors cite their own prior work on texture bias but omit the foundational *Geirhos et al., 2019 ICLR* paper, which is essential to contextualize this topic. Key references must be included.

7. The authors should consider discussing recent findings (e.g., O’Connell et al., 2025, *Open Mind*) showing that multi-view representations—not simply hierarchical abstraction—may explain improved model–human alignment in 3D perception.

Minor comments:

1. Why is *point density* treated as a within-subject factor while *orientation* is across-subject? This design choice should be justified.

2. The statement “The stimuli were selected from the test set… to ensure a fair comparison” is unclear—why does choosing from the test set guarantee fairness between humans and models?

3. typo in Fig 9: “scrabled” → “scrambled”

4. It would be informative to report the parameter counts for DGCNN and Point Transformer to clarify whether performance differences might stem from model capacity rather than architecture alone.

**Have the authors made all data and (if applicable) computational code underlying the findings in their manuscript fully available?**

The PLOS Data policy requires authors to make all data and code underlying the findings described in their manuscript fully available without restriction, with rare exception (please refer to the Data Availability Statement in the manuscript PDF file). The data and code should be provided as part of the manuscript or its supporting information, or deposited to a public repository. For example, in addition to summary statistics, the data points behind means, medians and variance measures should be available. If there are restrictions on publicly sharing data or code —e.g. participant privacy or use of data from a third party—those must be specified.requires authors to make all data and code underlying the findings described in their manuscript fully available without restriction, with rare exception (please refer to the Data Availability Statement in the manuscript PDF file). The data and code should be provided as part of the manuscript or its supporting information, or deposited to a public repository. For example, in addition to summary statistics, the data points behind means, medians and variance measures should be available. If there are restrictions on publicly sharing data or code —e.g. participant privacy or use of data from a third party—those must be specified.

Reviewer #1: Yes

Reviewer #2: Yes

Reviewer #3: **No:**

PLOS authors have the option to publish the peer review history of their article (what does this mean? ). If published, this will include your full peer review and any attached files.). If published, this will include your full peer review and any attached files.

**Do you want your identity to be public for this peer review?** For information about this choice, including consent withdrawal, please see our For information about this choice, including consent withdrawal, please see our Privacy Policy ..

Reviewer #1: No

Reviewer #2: No

Reviewer #3: **Yes:** Zitong LuZitong Lu

**Figure resubmission:**
---

## [Decision Letter · Decision Letter 1]

20 Feb 2026

Dear Dr. Fu,

We are pleased to inform you that your manuscript 'Hierarchical abstraction drives human-like 3-D shape processing in deep learning models' has been provisionally accepted for publication in PLOS Computational Biology.

Best regards,

Jian Liu

Academic Editor

PLOS Computational Biology

Marieke van Vugt

Section Editor

PLOS Computational Biology

There are a few minor concerns that, if addressed, would help finalize the manuscript for publication.

Reviewer's Responses to Questions

**Comments to the Authors:**

Reviewer #1: I would like to thank the authors for addressing my questions in the first round of the review. My questions were largely about technical details, motivation, and interpretation of the experimental design and results. I am satisfied with the authors’ response, especially with narrowing the conclusions that were overly claimed to be general. I also agree that the writing of the experimental sections has improved. However, while reading the other reviewers’ comments and the revised manuscript, I still have the following two concerns:

1. I appreciate that the authors rewrote the abstract and introduction. However, the new version does not read better than before. In the abstract, “We hypothesize that training with about 10k object instances enables models to form representations of local geometric structures in 3D shapes. However, their representations of global 3D object shapes are still limited. To test this hypothesis, we conducted three human experiments systematically…” the logic is broken. First, the hypothesis does not feel like a proper scientific hypothesis. Second, the hypothesis is about the model; it is unclear why human subjects are necessary to test it.

The new version emphasizes the “representation of 3D object shape.” Conventionally, “representation” often indicates that the subject of study is internal neural activation patterns. However, this study is entirely about behavioral-level comparison. The confusion matrix may hint at some information about the “representation,” but this study cannot answer questions like “…whether they acquire 3D object-shape representations similar to humans…”. Reviewer #2 mentioned RSA analysis, which is one type of representational analysis. Since the human-subject experiments are behavioral only, it is unlikely that such analysis, or any representation-based analysis, can be used.

2. Reviewer #3 mentioned the scope of this study. I agree that this study provides very limited insights into understanding biological vision. The authors argue that “this is the first study to employ point cloud stimuli across a broad range of object categories and a large number of object instances.” I find this argument weak. It is unclear why we would want to use point clouds as a vision-study paradigm, despite their popularity in computer-vision studies and applications.

Reviewer #2: The author addressed my concerns, and I recommended that the manuscript be accepted.

Reviewer #3: I am satisfied with this version.

**Have the authors made all data and (if applicable) computational code underlying the findings in their manuscript fully available?**

The PLOS Data policy requires authors to make all data and code underlying the findings described in their manuscript fully available without restriction, with rare exception (please refer to the Data Availability Statement in the manuscript PDF file). The data and code should be provided as part of the manuscript or its supporting information, or deposited to a public repository. For example, in addition to summary statistics, the data points behind means, medians and variance measures should be available. If there are restrictions on publicly sharing data or code —e.g. participant privacy or use of data from a third party—those must be specified.requires authors to make all data and code underlying the findings described in their manuscript fully available without restriction, with rare exception (please refer to the Data Availability Statement in the manuscript PDF file). The data and code should be provided as part of the manuscript or its supporting information, or deposited to a public repository. For example, in addition to summary statistics, the data points behind means, medians and variance measures should be available. If there are restrictions on publicly sharing data or code —e.g. participant privacy or use of data from a third party—those must be specified.

Reviewer #1: Yes

Reviewer #2: Yes

Reviewer #3: Yes

PLOS authors have the option to publish the peer review history of their article (what does this mean? ). If published, this will include your full peer review and any attached files.). If published, this will include your full peer review and any attached files.

**Do you want your identity to be public for this peer review?** For information about this choice, including consent withdrawal, please see our For information about this choice, including consent withdrawal, please see our Privacy Policy ..

Reviewer #1: **Yes:** Xu PanXu Pan

Reviewer #2: No

Reviewer #3: **Yes:** Zitong LuZitong Lu

---

## [Editor Report · Acceptance letter]

PCOMPBIOL-D-25-01916R1

Hierarchical abstraction drives human-like 3-D shape processing in deep learning models

Dear Dr Fu,

I am pleased to inform you that your manuscript has been formally accepted for publication in PLOS Computational Biology. Your manuscript is now with our production department and you will be notified of the publication date in due course.

With kind regards,

Anita Estes
